# Towards Sustainable Early Education Practices: A Quasi-Experimental Study on the Effects of Kindergarten Physical Education Programs on Fundamental Movement Skills and Self-Regulation in Haikou City, China

**Hua Wu [1,2]**, **Wichai Eungpinichpong [3,\*]**, **Hui Ruan [2,\*]**, **Wenke Chen [2]**, **Yibei Yang [2]** and **Xiujuan Dong [2]**

[1] Faculty of Associated Medical Sciences, Khon Kaen University, Khon Kaen 40002, Thailand; wuhua0049@gmail.com

[2] Faculty of Physical Education, Hainan Normal University, Haikou 571158, China; chenwenke0513@163.com (W.C.); chbhnyy0828@163.com (Y.Y.); m18789256920@163.com (X.D.)

[3] BNOJHP Research Center, PT Division of Physical Therapy, Faculty of Associated Medical Sciences, Khon Kaen University, Khon Kaen 40002, Thailand

\* Correspondence: wiceun@gmail.com (W.E.); huiruan@kkumail.com (H.R.)

**Abstract:** Acquiring fundamental movement skills (FMS) in early childhood is linked to long-term engagement in physical activity, promoting lifelong health. Concurrently, the development of self-regulation contributes to fostering socially responsible and empathetic individuals. This study aims to contribute to the sustainable development goals of Good Health and Well-being (Goal 3) and Quality Education (Goal 4) by promoting early education practices that enhance children's physical and socio-emotional development. This quasi-experimental study, employing a pre/post-test control design, examined the influence of different kindergarten physical education programs on FMS and self-regulation. Participants from a sports-themed kindergarten, ordinary public kindergarten, and non-profit private kindergarten were equally divided into three groups. The "Hello Sunshine" (HS) group followed a ball game-based program; the ordinary physical education (OPE) group's curriculum had multi-themed physical activities; and the free-play (FP) group as a comparison group (free play) spent their activity time freely playing. Data from a total of 239 children were analyzed, all of whom received identical assessments of fundamental movement skills and self-regulation by the Test of Gross Motor Development-3 (TGMD-3) and the head–toes–knees–shoulders (HTKS) test at baseline and after 10 weeks of the PE curriculum. There was a significant effect of grouping on TGMD-3 composite scores after controlling for the pre-test score (F(2, 235) = 65.232, $p < 0.001$, Partial $\eta^2$ = 0.357). The composite score of the HS group was clearly higher than that of the OPE (95% CI:10.72~16.45) and FP (95% CI: 4.16~9.98) groups. A significant group time effect was observed for self-regulation (F(2, 236) = 4.588, $p = 0.011$, Partial $\eta^2$ = 0.037). After 10 weeks, the HS group displayed a more significant increase in self-regulation (14.8%) than that in the OPE (9.7%) and FP (14.6%) groups. The ball game-based program exhibited more advantages in promoting fundamental movement skills and self-regulation among the kindergarteners. Overall, this study's findings highlight the potential benefits of kindergarten physical education programs and underscore the importance of early childhood development, emphasizing its potential to contribute to holistic child development and align with sustainability goals.

**Keywords:** motor competence; cognition; ball skills; young children; physical education

## 1. Background

From the very beginning of life, early childhood care and education serve as the cornerstone for the sustained development, well-being, and health of children. Early childhood is a crucial stage in the development of children's physical, motor, emotional, and social abilities [1]. Sustainable Development Goal 4 within Education 2030 emphasizes the need for

universal access to quality early childhood development, care, and pre-primary education, aiming at fostering holistic children's development, and ensuring that all children acquire the knowledge and skills needed to promote sustainable development [2]. However, most preschool children currently face at least three problems that pose a negative, long-term impact on their development: (1) physical inactivity [3], (2) poor fundamental movement skills (FMS) [4,5], and (3) inadequate preparation for school [6]. Meanwhile, childhood obesity and its associated health issues have garnered international attention. The World Health Organization (WHO) reported that an estimated 38.2 million children under five years of age were overweight or obese in 2019, especially in developing countries, and the prevalence rate is increasing rapidly [7]. In particular, the COVID-19 pandemic has seen forced closures of schools and parks, people isolated at home for weeks and months, reductions in physical activity, and increases in sedentary behavior, with studies reporting limitations in children's physical fitness and motor skill development [8,9], and these changes may have even had long-lasting consequences [10,11]. Therefore, these problems urgently need to be improved in the post-pandemic era.

FMSs consist of basic movements that form the foundation for more complex and specialized physical activities and comprise three components: locomotor skills, object control skills, and stability skills [12,13]. One review synthesizes the relationship between FMS competency and eight potential benefits (i.e., global self-concept, perceived physical competence, and reduced sedentary behavior) in children and adults [14]. FMS at preschool age is a predictor of adolescent and adult physical activity (PA) [15,16]. However, the systematic review found that global levels of FMS in children only correspond to "below average" to "average" TGMD-2 normative data [17]. Therefore, the crucial implementation of fundamental movement skills (FMS) interventions in early education environments aligns with Sustainable Development Goal 4 (Quality Education) by ensuring that children develop holistically, acquiring the physical and cognitive skills necessary for a sustainable and prosperous future. Moreover, the cultivation of FMS in early childhood correlates with Sustainable Development Goal 3 (Good Health and Well-being), fostering a lasting dedication to physical activity and overall well-being [18].

The preschool years are critical to the development of children's FMS, as supported by motor development theory [19]. However, certain children's FMS, such as specific throwing or kicking techniques, are not learned naturally through free play and may require intentional practice and instruction [20]. Per Zeng's [21] systematic review, of the 10 studies assessing the effects of PA on motor skills, 8 report significant improvements in FMS, and 1 observed mixed findings. Moreover, one failed to promote any beneficial outcomes.

Further, the development of motor and cognitive functions is deeply interconnected to infancy and early childhood. There is increasing evidence of a positive association between FMS, PA, and self-regulation [21–23]. Self-regulation is a multi-dimensional concept, which is a higher-order cognitive process that usually refers to the ability to control or direct one's attention, thoughts, emotions, and actions [24]. Effective self-regulation, including but not limited to concentrating on tasks, relying on the working memory function to extract established behavioral norms, utilizing inhibitory control to restrain instinctual reactions, and accessing existing behavioral norms to take final actions, is highly relevant to positive classroom behavior and academic achievement [25–27]. In the Head Start Early Learning Outcomes Framework, self-regulation is acknowledged as a pivotal component in the early development of children [28]. Furthermore, in the context of sustainability, individuals with refined self-regulation skills are well positioned to engage in socially responsible behaviors, contributing to the development of a sustainable and conscientious global community [29]. Studies find that self-regulation ability develops rapidly in early childhood and is affected by many ecological factors, such as individual development and traits, parenting, and early childhood education [30–32]. Many studies focus on designing classroom or academically relevant interventions for socioeconomically disadvantaged children, aiming to improve their behavioral problems and enhance school readiness, such as the Head Start REDI

program [33], Chicago School Readiness Project [34], and Tools [35], but there is limited research on intervening self-regulation through exercise.

However, evidence indicates that PA contributes to improved cognitive abilities [36]. van der Fels et al. find a specific association between physical activity and certain aspects of cognitive performance and suggest complex motor intervention programs to promote both motor and higher-order cognitive skills in pre-adolescent children [37]. This provides a basis for using PA interventions to promote changes in self-regulation. A recent study by Robinson et al. finds that the Children's Health Activity Motor Program (CHAMP) can enhance self-regulation [22]. Miller et al. show that CHAMP significantly improved behavioral self-regulation and had no effect on cognitive self-regulation [38]. In another study, games and exercise improved partial self-regulation [39], and other relevant studies using comprehensive exercise interventions report mixed results [40–43]. These inconsistent results may be related to differences in study design and the self-regulation measurement. Our study utilized the head–toes–knees–shoulders (HTKS) task to test behavior self-regulation, which has been demonstrated to have high reliability and ecological validity. The task evaluates inhibitory control and the ability to follow specific instructions and inhibit impulsive responses. It requires children to listen to instructions and respond with large muscle movements, as it reflects control and directive behavior in the classroom [27].

In China, the vast majority of children spend more than eight hours each day in daycare, where kindergarten PA policy, outdoor activity time, and teacher PA strategy influence their PA [44,45]. Kindergartens can provide a wide range of public health benefits by serving as a key intervention environment for PA. For example, providing organized and structured teaching activities to promote fitness and skill development in children, outdoor activities, including engaging in activities and playing in outdoor environments, and nature-based education, offer benefits such as enhanced physical health, cognitive development, and emotional well-being [46]. However, challenges like safety concerns, weather dependency, and resource constraints must be addressed to ensure successful and sustainable implementation, fostering a holistic learning environment for children. In recent years, the Chinese government has proposed many policies and initiatives aimed at improving physical education (PE) for young children. However, no comparative study has been conducted on the effects of mainstream PE curricula in Chinese kindergartens.

This study aimed to compare the effects of two common groups of PE programs with a control group (free-play activities) in Chinese kindergartens, including a ball-themed program, regular PE courses, and free play, on children's FMS and self-regulation. We hypothesized that the FMS scores of both two groups of preschoolers have significantly improved and that the HS group would have the greatest improvement. Furthermore, we hypothesized that the two groups of children would significantly improve self-regulation, especially in the HS group.

## 2. Materials and Methods

This quasi-experimental study was conducted in three kindergartens (sports-themed public kindergarten, ordinary public kindergarten, and non-profit private kindergarten) in the center of Haikou City, China. The trial was approved by the Khon Kaen University Ethics Committee for Human Research (HE642100), registered with the Chinese Clinical Trial Registry (registration number: ChiCTR2000035414) on 8 October 2020, and conducted in accordance with the Declaration of Helsinki.

### 2.1. Participants

This study's sample size was determined using WinPepi software (Version 11.65) based on Westendorp's research [47], which indicated that 83 participants in each kindergarten would be sufficient for a significance level of 5%, a power level of 80%, and a confidence level of 95%, inflated to compensate for a 5% drop-out rate to test the primary outcome (e.g., FMS). A total of 249 children ($5.48 \pm 0.59$ years old; 46% girls; no ethnic minorities) were enrolled in this study. Each kindergarten class was considered an experimental

unit, with children from two classes each of grades K2 and K3 selected. Only healthy and typically developing children, aged 4–6 years, whose primary caregivers provided informed consent, were eligible to participate. Children who (1) had communication difficulties or unexplained health problems, (2) had contraindications to exercise, such as congenital heart disease or asthma, and (3) had additional training from a sports club were excluded.

*2.2. Procedures*

The kindergarten students were recruited at the beginning of the school term (March 2022). The teacher placed an informed consent form in the students' backpacks for parents to consider. Researchers and teachers screened the children who returned the consent form. Baseline data, including height, weight, FMS, and self-regulation, of the enrolled children were collected in the first week. The FMS test was conducted in a sports field, while the self-regulation test was held in a separate classroom. With the cooperation of the teachers and based on the time and space arrangements of each kindergarten, all data were collected within two weeks. If a child was absent on the day of the test, data were collected on another day to minimize missing data. Each child received a sticker or snack as a reward after completing the test. After 10 weeks of PE curricula, the post-test data were collected. The study procedure is shown in Figure 1.

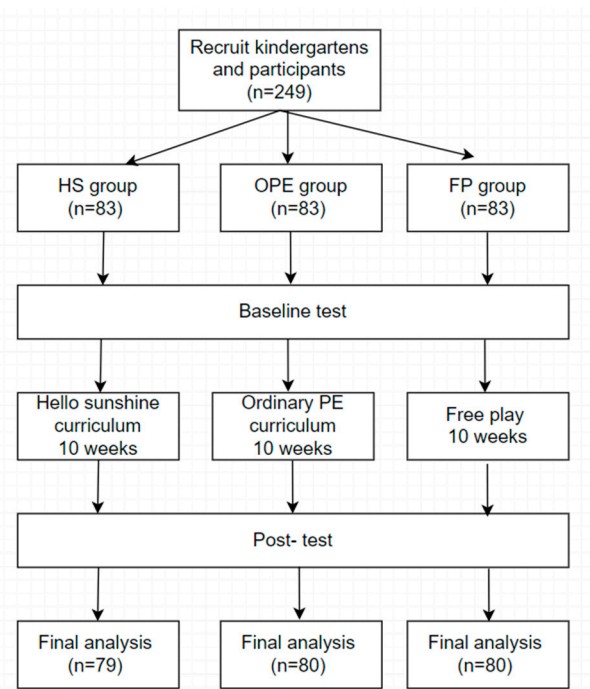

**Figure 1.** Study procedure.

*2.3. Intervention*

Kindergartens operate on a five-day school week. Children in the sports-themed kindergarten received the ball game-based program "Hello Sunshine" (HS group). The ordinary public kindergarten followed a theme-based program for daily life (OPE group), while the non-profit private kindergarten allowed children free play during activity time (FP group). Apart from the PE classes and outdoor activities, the daily activities in each kindergarten remained the same. In the HS and OPE groups, PE classes in grade K2 were held once a week, lasting 25 min each session. In grade K3, PE classes were held twice a week, with each session lasting 30 min. Outdoor activities were scheduled for 60 min, five times a week. The FP group had one hour of outdoor activity every day. All three groups participated in a 10-week PE intervention. The PE program for the HS and OPE groups

totaled 54 h in K2 and 60 h in K3, while the FP group's PE totaled 50 h. Additional details are described below.

### 2.3.1. HS Group

'Hello Sunshine' is an upgraded version of this kindergarten ball-based characteristic curriculum, which is based on the 3–6 Years Age Early Learning and Development Guideline of our country [48] and the comprehensive Seefeldt's motor development theory [49]. It has been meticulously designed by a team of PE teachers led by the principal, taking into consideration the age and psychological characteristics of young children. The implementation of children's ball activities is approached from the perspective of the children, adhering to the principles of playfulness and appropriateness. The ball game-based program incorporated an array of balls like football, basketball, badminton, ping-pong, table tennis, baseball, golf, handball, and bowling. The program focused on developing various ball skills, including kicking, dribbling, hitting, et al. Considering program design, Newell's constraints model [50] served as the theoretical foundation. First, the ecological task analysis method was applied to consider two or three critical individual constraints related to the motor tasks. Second, environmental or task constraints that could be manipulated were identified to control the difficulty of the motor tasks. Finally, the range of task difficulty was determined based on the developmental level of the children's motor skills [51]. For example, when learning the badminton forehand stroke, hand-eye coordination and strength are individual constraints. Adjustable task constraints include hitting a stationary ball (by tying and suspending the ball with a string) and hitting a moving ball (self-tossed). The height and speed of hitting represent another task constraint, which is related to strength. Environmental constraint adjustment involves using a short-handled plastic badminton racket, replacing feathers with balloons, and using storytelling to stimulate children's enthusiasm for hitting. Finally, teachers adjust the difficulty level based on the individual abilities of the children. The structure of each class includes a 5 min warm-up period, a 20–25 min exercise period, and a 5 min cool-down activity. There was a high degree of autonomy during outdoor time. In addition to self-selected ball skill activities once (K2) or twice (K3) each week, participants could choose their area of exercise, such as the equipment zone or bike zone, during the remainder of outdoor time (Table 1).

**Table 1.** Example of weekly PE program content.

| | Introduction | Preparation | Main Part of the Lesson | Game | Final Part of the Lesson | Outdoor Activity |
|---|---|---|---|---|---|---|
| | | | HS Group | | | |
| K2 | Animal fun running | General preparatory exercises with footballs | Explore dribbling and ball control | Relay play obstacle run | Body tapping and stretching with music | Self-selected ball skills with PE teacher |
| K3 (1st) | Sliding in various directions | Fancy dribbling with basketballs | Basketballs are tossed and caught in pairs | Catch basketballs for different tasks | Body tapping and stretching with music | Self-selected ball skills with PE teacher |
| K3 (2nd) | Running through obstacles | Simple toss and catch with badminton | Forehand stroke, fixed ball with short-handled plastic badminton rackets | Forehand stroke with specific objective | Body tapping and stretching with music | Self-selected zone |
| | | | OPE Group | | | |
| K2 | Imitating animals | Climbing and rolling exercises | Practice climbing and rolling | Tire-pushing race | Body tapping and stretching with music | Self-selected zone |
| K3 (1st) | Running in a circle | Practice rope swinging | Jumping rope | Explore ways to play with rope | Body tapping and stretching with music | Self-selected zone |

**Table 1.** *Cont.*

|  | Introduction | Preparation | Main Part of the Lesson | Game | Final Part of the Lesson | Outdoor Activity |
|---|---|---|---|---|---|---|
| K3 (2nd) | Running in a circle | Body balance exercises | Practice the combo jungle gym | Relay play with crawling | Body tapping and stretching with music | Self-selected zone |
| FP Group | | | | | | |
| Participants used various playground facilities (such as slides and sandboxes) to play freely | | | | | | |

2.3.2. OPE Group

PE classes were taught by a PE teacher who referred to a specified kindergarten textbook for various monthly themes, including knowledge and activities related to health and PA. In these 10 weeks, themes included "Transportation Exhibition" and "My Friends" in grade K2 and "Change" and "I love . . ." in K3. Basic skills, such as walking, running, rolling, crawling, jumping, and throwing, were taught through the themes. Outdoor time was the same as that of the HS group, but children were allowed to freely play in different areas.

2.3.3. FP Group

Children are given the autonomy to choose playmates and activities based on their personal interests, with an emphasis on child-led activities. The playgrounds, which are equipped with swings, slides, sandboxes, and open spaces, serve as venues for physical activities. Despite lacking a PE background, teachers focus on supervision to ensure a safe environment. They refrain from actively guiding or directing children's play unless safety concerns arise.

*2.4. Outcome Measures*

2.4.1. Demographic and Anthropometric Data

The demographic information of the children (such as date of birth and sex) was provided by the teachers, and their height and weight were measured using an electronic scale (height and weight tester, HW-900B, Zhengzhou, China) on the morning of the assessment conducted by the research assistant.

2.4.2. FMS

The Test of Gross Motor Development-3 (TGMD-3), which is an effective tool for measuring the gross motor skills of children aged 3–10 years, was used for FMS assessment [52,53]. The TGMD-3 has demonstrated good internal consistency reliability, test–retest reliability, and construct validity [52,54]. TGMD-3 is divided into two subscales: locomotor skills (run, gallop, hop, leap, jump, and slide) and ball skills (two-hand strike, one-hand strike, dribble, kick, catch, overhand throw, and underhand throw). The TGMD-3 was conducted with groups of 4–5 participants, with each group session lasting approximately 30 min. A trained administrator demonstrated each skill's correct movements. Then, each participant performed one practice trial followed by two formal tests. An experienced scorer reviewed the recorded videos of these tests. According to the testing criteria, each skill was assessed on the basis of three to five performance criteria. If a child demonstrated the correct performance criteria, the child received a score of one for each criterion; otherwise, a score of zero was given. Raw scores were calculated by adding the scores of the two formal tests.

2.4.3. Self-Regulation

The head–toe–knees–shoulders (HTKS) test evaluates children's behavioral self-regulation, has strong inter-rater reliability [25,26], and is a good assessment tool for behavioral self-regulation in 4–8-year-old children regardless of their cultural background [24].

The children were individually tested by professional psychologists and their assistants in separate rooms within the kindergarten. The test is divided into two tasks. The first is the head-to-toe task, in which the child was asked to respond in the opposite way to the examiner's two verbal instructions. The second task involved mixed knee and shoulder commands, in which the child should also respond in the opposite manner to the commands [55]. The HTKS test consists of 26 irregular commands, and the child received two points for a correct response, zero points for an incorrect response, and one point if an incorrect response was immediately corrected. The maximum score is 52 [56]. This test has been used in research on Chinese preschool children [57].

### 2.5. Data Analysis

Statistical analysis data were analyzed using SPSS (version 25.0; IBM, Armonk, NJ, USA). The normality of the data was verified using the Kolmogorov–Smirnov test and further descriptive statistics were performed. Intra-group comparisons were conducted using paired t-tests or Wilcoxon paired-sample tests depending on the distribution of the data. The baselines were used as a co-variable and a covariance analysis was performed in comparison to the TGMD-3 results. The 3 (HS, OPE, and FP) $\times$ 2 (baseline and after 10 weeks) mixed-model ANOVA comparison was used in the comparison of the HTKS results. The effect size was expressed as partial $\eta 2$ ($\eta p2$) and was defined as small (0.01), medium (0.06), and large (0.14).

## 3. Results

Ten preschool children had missing or outlier data for one or more outcome variables either at baseline or after 10 weeks, resulting in their exclusion from the analysis. The final sample consisted of 239 kindergarten children (mean age: $5.49 \pm 0.60$ years; 54% boys). Detailed demographic and anthropometric information are provided in Table 2.

**Table 2.** Demographic characteristics.

|  | HS Group (*n* = 79) | OPE Group (*n* = 80) | FP Group (*n* = 80) | Total (*n* = 239) |
|---|---|---|---|---|
| Sex |  |  |  |  |
| Boy | 40 (50.63%) | 43 (53.75%) | 47 (58.75%) | 130 (54.39%) |
| Girl | 39 (49.37%) | 37 (46.25%) | 33 (41.25%) | 109 (45.61%) |
| Age | $5.38 \pm 0.52$ | $5.48 \pm 0.67$ | $5.59 \pm 0.58$ | $5.49 \pm 0.60$ |
| Height | $112.9 \pm 6.46$ | $117.68 \pm 8.28$ * | $112.64 \pm 5.42$ | $114.41 \pm 7.18$ |
| Weight | $19.94 \pm 3.79$ | $20.59 \pm 4.07$ | $19.63 \pm 2.92$ | $20.05 \pm 3.63$ |
| BMI (kg/m$^2$) | $15.65 \pm 2.29$ | $15.39 \pm 2.01$ | $15.49 \pm 2.22$ | $15.51 \pm 2.17$ |

* Note: $p < 0.05$ (Significant Difference).

### 3.1. FMS

When comparing the baseline TGMD-3 scores of the three groups, we found significant differences in both the locomotor ($F_{(2, 236)} = 8.651$, $p < 0.001$) and composite scores ($F_{(2, 236)} = 4.115$, $p = 0.018$). However, ball skill scores were similar across the groups ($F_{(2, 236)} = 2.735$, $p = 0.067$).

After 10 weeks, the TGMD-3 scores in all three groups improved compared to those in the pre-test. When the baseline was controlled, the independent variable (grouping) showed a significant effect; the difference in the TGMD-3 subset and composite score between the groups is presented in Table 3. The results of the pairwise comparisons are presented in Table 4.

**Table 3.** TGMD-3 scores among the groups, baseline adjusted using ANCOVA.

| Variable | Group (Intervention) | Baseline | | After Intervention (Adjusted) | | *p*-Value | Effect Size |
|---|---|---|---|---|---|---|---|
| | | Mean | SD | Mean | SD | | |
| Locomotor skills score | HS | 22.99 | 4.98 | 28.08 | 4.80 | 0.001 * | 0.058 |
| | OPE | 20.51 | 5.72 | 27.38 | 4.92 | | |
| | FP | 23.71 | 4.56 | 25.29 | 4.83 | | |
| Ball skills score | HS | 14.16 | 4.71 | 28.60 | 5.19 | <0.001 ** | 0.424 |
| | OPE | 13.30 | 4.65 | 21.87 | 5.10 | | |
| | FP | 12.44 | 4.65 | 17.86 | 5.19 | | |
| TGMD composite score | HS | 37.15 | 7.47 | 56.569 | 7.54 | <0.001 ** | 0.357 |
| | OPE | 33.81 | 7.78 | 49.502 | 7.57 | | |
| | FP | 36.15 | 7.42 | 42.986 | 7.50 | | |

Note: * $p < 0.05$ (Significant Difference), ** $p < 0.001$ (Significant Difference). TGMD-3: Test of Gross Motor Development-3; HS: Hello Sunshine; OPE: ordinary physical education; FP: Free Play.

**Table 4.** Pairwise comparison of TGMD-3 scores among the groups.

| Group | | Locomotor Skills Score | | Ball Skills Score | | TGMD-3 Composite Score | |
|---|---|---|---|---|---|---|---|
| | | *p* | Difference (95% CI) | *p* | Difference (95% CI) | *p* | Difference (95% CI) |
| HS | FP | 0.001 ** | 0.97~4.65 | <0.001 ** | 8.74~12.72 | <0.001 ** | 10.72~16.45 |
| | OPE | 1 | −1.18~2.57 | <0.001 ** | 4.75~8.69 | <0.001 ** | 4.16~9.98 |
| OPE | FP | 0.023 * | 0.22~4.00 | <0.001 ** | 2.05~5.97 | <0.001 ** | 3.64~9.39 |
| | HS | 1 | −2.57~1.18 | <0.001 ** | −8.69~−4.75 | <0.001 ** | −9.98~−4.16 |
| FP | OPE | 0.023 * | −4.00~−0.22 | <0.001 ** | −5.97~−2.05 | <0.001 ** | −9.39~−3.64 |
| | HS | 0.001 ** | −4.65~−0.97 | <0.001 ** | −12.72~−8.74 | <0.001 ** | −16.45~−10.72 |

Note: * $p < 0.05$ (Significant Difference), ** $p < 0.001$ (Significant Difference). TGMD-3: Test of Gross Motor Development-3; HS: Hello Sunshine; OPE: ordinary physical education; FP: Free Play.

### 3.2. Self-Regulation

The Wilcoxon paired-sample test revealed that the FP group significantly increased their Part 1 (Z = −6.953, $p < 0.001$), Part 2 (Z = −6.213, $p < 0.001$), and total (Z = −7.617, $p < 0.001$) scores. The OPE group showed a similar upward trend in their Part 1 (Z = −6.227, $p < 0.001$), Part 2 (Z = −3.594, $p < 0.001$), and total (Z = −7.055, $p < 0.001$) scores. The HS group also showed significantly better Part 1 (Z = −3.757, $p < 0.001$), Part 2 (Z = −6.254, $p < 0.001$), and total (Z = −6.748, $p < 0.001$) scores. There was no significant difference in HTKS scores among the three groups at baseline (F(2, 236) = 1.357, $p = 0.259 > 0.05$). The mixed-model ANOVA was used to assess the effect of different groupings and time. While the main effect of school was not significant (F(2, 236) = 1.652, $p = 0.194$), the interaction between time and school was ($p = 0.011$) (Table 5).

**Table 5.** Comparison of HTKS scores among the groups.

| Variable | | | | | A Group-by-Time Interaction Effect Size | | |
|---|---|---|---|---|---|---|---|
| HTKS Score | Pre-Test Mean ± SD | Post-Test Mean ± SD | % Δ | *p*-Value within Group between Time | F | *p* | Partial η² |
| HS | 34.73 ± 8.61 | 39.86 ± 6.74 | 14.8 | <0.001 ** | 4.588 | 0.011 ** | 0.037 |
| OPE | 34.15 ± 8.59 | 37.46 ± 7.82 | 9.7 | <0.001 ** | | | |
| FP | 32.31 ± 11.54 | 37.03 ± 11.69 | 14.6 | <0.001 ** | | | |

Note: ** $p < 0.001$ (Significant Difference). HTKS: Head–Toes–Knees–Shoulders; HS: Hello Sunshine; OPE: ordinary physical education; FP: Free Play.

## 4. Discussion

This quasi-experimental study explored the effects of two different kinds of PE programs with comparison (free play) on children's motor skills and self-regulation development. All three groups demonstrated significant improvements in the FMS and self-regulation tests compared to baseline levels. In the FMS sub-test, especially for ball skills, the HS group performed better than the OPE and PE groups and exhibited a large effect size. Both the HS and FP groups revealed more significant progress in their HTKS test results than the OPE group. Overall, our results indicate that regular PA, particularly ball skill-based activity, contributes to the development of motor skills in kindergarteners and promotes the development of self-regulation abilities.

### 4.1. Comparison of FMS Changes in the Three Groups

Gao et al. [58] have concluded that interventions during the preschool stage are crucial for the development of FMS. They observed significant improvements in FMS following an intervention lasting only eight weeks. In terms of the impact of PE intervention on FMS, the observed results were in accordance with several previous studies, which established that structured PA intervention not only increases children's PA level. For example, Palmer et al.'s study compared the physical activity participation of preschool children in two different physical activity opportunities: unstructured outdoor play and structured movement classes. The study found that structured movement time provides children with more opportunities to engage in physical activity compared to outdoor time alone [59]. Similarly, Tortella et al. found that, compared to the free-play group and the control group, the partially structured activity group showed significant improvements in specific tests measuring motor skills [60]. Furthermore, they discovered that this group effectively improved and maintained their motor skills better than free play [61,62].

Experts agree that FMS requires guided teaching and learning, followed by practice and reinforcement before they can be significantly improved [13,63]. Results from a meta-analysis study [63] indicate that FMS significantly improved pre- to post intervention, but the control group's (i.e., free play) improvement was not significant. The HS program envisages learning and repeatedly practicing various basic ball skills such as dribbling, kicking, holding, and tapping the ball. These skills provide structured and guided instruction, allowing preschoolers to learn and refine their motor skills in a targeted manner. By engaging in ball-related activities, they can improve their coordination, balance, and manipulative skills, which are essential components of FMS. Our results are consistent with the study of Robinson et al. [20], which used ball skills intervention findings to suggest that providing a high-quality motor skill program in early childhood settings could potentially be a sustainable public health approach for promoting FMS and positive physical developmental trajectories for health [20]. Conversely, free play is characterized by unstructured and child-initiated activities, allowing preschoolers to explore and experiment with a wide range of movements and play scenarios. While free play may not provide explicit instruction on specific skills, it offers opportunities for spontaneous and self-directed practice of FMS. Children engage in imaginative play, create their own rules, and adapt their movements to different situations, which can contribute to the development and refinement of FMS. On the whole, our results are supported by previous studies both in theory and practice.

### 4.2. Comparison of HTKS Changes in the Three Groups

The results showed that PE intervention improved self-regulation in kindergartners. Among the three groups, the HS and FP groups showed a greater increase in self-regulation scores. Our findings are supported by previous related studies. As an acute intervention during classroom movement breaks, Ureña et al. [64] conducted a 15 min exercise program with different levels of difficulty for 49 children aged 4–5 years. Regardless of difficulty, the program showed a certain intervention effect on self-regulation. Using a series of music and movement games can also improve self-regulation [65,66]. Duman et al. [67] adopted embedded learning-based movement education for 5-year-old children, whose

activities—group activities, cooperative games, and social interaction—required gross motor, fine motor, and visuomotor coordination skills, and they also found that the experiment group improved self-regulation. Korll et al. [68] observe a strong interaction among the program, environment, daily activities, and development of self-regulation, suggesting the need to include play in the kindergarten curriculum, in line with our ball game-based curriculum design.

Several possible mechanisms accounted for FMS, and self-regulation improved in our study. From a neuroscientific perspective, shared cerebellar and/or prefrontal processes might be involved in the relationship between motor function and cognition [47]. Research suggests that various exercises affect different neurocognitive networks [69,70]; the cerebellum is important for complex and coordinated movements, and the prefrontal cortex is critical for higher-order cognitive functioning [71]. Engaging in more complex movement patterns requires deeper information processing, leading to more consistent neural plasticity changes [72]. Furthermore, the development timeline of motor skills and executive function is similar, with accelerated development between the ages of 5 and 10.

It is important to note that the specific impact of these programs may vary depending on the design, duration, intensity, and quality of the interventions, as well as individual differences among children. Nonetheless, incorporating ball skill-based programs, theme-based PE, and free-play content into early childhood settings can offer valuable opportunities for FMS development and the cultivation of self-regulation skills in preschoolers.

The strengths of this study were two fold. First, it examined the groups of subjects, whose PA patterns represented the main forms of PE in existing Chinese kindergartens. Our results provide a theoretical basis and practical reference for the effectiveness of PE for young children, which has received scant attention in the current research. Second, our study covers the developmental evaluation of motor and cognitive domains, which highlights the important fields of children's development.

A potential weakness of this study was that it considered only the effects of program intervention, without considering the possible effects of environmental factors such as kindergarten teachers, facilities, and playgrounds on the outcome measures. The exclusion of these environmental factors could limit the comprehensive understanding of the observed outcomes, as they play crucial roles in shaping the overall educational experience for children in a kindergarten setting. Future research should consider exploring these environmental variables to provide a more holistic assessment of the intervention's effectiveness. Another limitation was the use of a subjective measurement instrument, including the assessment with TGMD-3, and the monitoring of physical activity intensity during interventions, which may be influenced by individual differences among observers.

## 5. Conclusions

This quasi-experimental study reveals that a ball game-based PE program is more effective in enhancing Fundamental Motor Skills (FMS) and self-regulation in kindergarten children compared to free play. The significance lies in emphasizing the potential benefits of structured PE interventions for early childhood development. Future research needs to explore the influence of teachers, facilities, and playgrounds on outcomes. Additionally, examining motor skill performance in diverse settings and establishing a comprehensive framework for acquisition can offer nuanced insights. These studies are essential to inform intervention guidelines and to promote sustainable development of children's motor skills and related areas.

**Author Contributions:** Conceptualization, H.W. and W.E.; Data Curation, H.R. and Y.Y.; Investigation, X.D.; Methodology, H.W. and Y.Y.; Resources, W.C.; Writing—Original Draft, H.W.; Writing—Review and Editing, W.E. All authors have read and agreed to the published version of the manuscript.

**Funding:** This research was funded by the Fundamental Fund of Khon Kaen University and has received funding support from the National Science, Research and Innovation Fund (NSRF),

Project No. SD-65-023, and the Hainan Province Philosophy and Social Science Planning Project (HNSK(YB)23-57).

**Institutional Review Board Statement:** The study was conducted in accordance with the Declaration of Helsinki, and approved by Khon Kaen University Ethics Committee for Human Research (HE642100, 9 September 2021).

**Informed Consent Statement:** Informed consent was obtained from all subjects involved in the study.

**Data Availability Statement:** The data are available from the corresponding author upon reasonable request.

**Acknowledgments:** The authors wish to thank all study participants.

**Conflicts of Interest:** The authors declare no conflicts of interest.

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
