# Peer review of "Towards Sustainable Early Education Practices: A Quasi-Experimental Study on the Effects of Kindergarten Physical Education Programs on Fundamental Movement Skills and Self-Regulation in Haikou City, China"

_sustainability, doi:10.3390/su16041400_

Round 1

Reviewer 1 Report

Comments and Suggestions for Authors

The manuscript is very interesting and cover a new field of knowledge bridging education, physical education and well-being, pre-school children  development. The text reads well, it is accurate and scientific sound. Nevertheless, it is also pleasant for a reader with limited knowledge in the field. Please see the further specific comments to the manuscript.

Title: add ‘from Haikou City, China’ at the end of the title, to specify the place of the case study.

Comments to lines:

56-57: Can you please expand more on this statement? This is also proven by your results.

47-48: Is there data from Covid 19 pandemic influences right after the data from 2019? Was there a worsening impact? it would add an interesting view

94: can you expand on ‘outdoor activity’, would this be related to outdoor education (and related field of nature-base ed, etc. too?). With which benefit and challenges?

131-132: please explain if there were any biased in the sample due to the parents’ signed consent form

152-153: please specify how many days a week the three groups have KG days (5, 6, 7?)

397: what about the children’s socioeconomic status? – check Rzavská, D.; Masaryková, D.; Antala, B. Sustainable Development of Basic Motor Competencies Related to Socioeconomic Status of Primary School Children. Sustainability 202214, 15175. https://doi.org/10.3390/su142215175

what about the environmental impact and contribution? like in Golding J, Emmett P, Iles-Caven Y, Steer C, Lingam R. A review of environmental contributions to childhood motor skills. J Child Neurol. 2014 Nov;29(11):1531-47. doi: 10.1177/0883073813507483. Epub 2013 Oct 29. PMID: 24170258; PMCID: PMC4004720.)?

404: please expand more on the conclusion, reporting the significance of the results of your research and highlight the need for further research that fills your research gaps.

Author Response

Response to Reviewer 1 Comments

Dear Reviewer

Thank you for your diligent review of our manuscript and for providing valuable feedback. We greatly appreciate your expertise, and your suggestions have played a crucial role in enhancing the quality of the paper.

In response to your questions and recommendations, we have made the necessary revisions and hope that these changes meet your expectations. Specifically, we modified in manuscript with track change and Note. We hope these modifications not only address your concerns but also contribute to an overall improvement in the paper.         

Point-by-point response to Comments and Suggestions for Authors

Comments 1: Title: add ‘from Haikou City, China’ at the end of the title, to specify the place of the case study.

Response 1: yes, add. Towards Sustainable Early Education Practices: A Quasi-Experimental Study on the Effects of Kindergarten Physical Education Programs on Fundamental Movement Skills and Self-Regulation from Haikou City, China

Comments 2: 56-57: Can you please expand more on this statement? This is also proven by your results.

Response 2: add

Comments 3: 47-48: Is there data from Covid 19 pandemic influences right after the data from 2019? Was there a worsening impact? it would add an interesting view

Response 3: Thank you. We add L84-90

In particular, the COVID-19 pandemic has seen forced closures of schools and parks, people isolated at home for weeks and months, reductions in physical activity and in-creases in sedentary behaviour, with studies reporting limitations in children's physi-cal fitness and motor skill development[8,9], and these changes may have even had long-lasting consequences[10,11]. Therefore, these problems urgently need to be im-proved in the post-pandemic era.

Comments 4: 94: can you expand on ‘outdoor activity’, would this be related to outdoor education (and related field of nature-base ed, etc. too?). With which benefit and challenges?

Response 4: Add L149-155 For example, provide organized and structured teaching activities to promotes fitness, skill development in children; outdoor activities, including engage in activities and play in outdoor environments, nature-based education, offer benefits such as enhanced physical health, cognitive development, and emotional well-being[43]. However, chal-lenges like safety concerns, weather dependency, and resource constraints must be ad-dressed to ensure successful and sustainable implementation, fostering a holistic learning environment for children

Comments 5: 131-132: please explain if there were any biased in the sample due to the parents’ signed consent form

Response 5: The method of placing consent forms in students' backpacks may introduce biases, as it relies on parents' voluntary participation, potentially excluding those from diverse socioeconomic, concept of parenting backgrounds. Such biases can impact the representativeness and generalizability of study findings.

Although most parents of children in China are very cooperative with teachers to complete their work, in order to reduce the risk of such bias, we should adopt more channels, such as direct delivery to parents,use class chat groups on WeChat and parent contact association to ensure a more representative sample.

Comments 6: 152-153: please specify how many days a week the three groups have KG days (5, 6, 7?)

Response 6: Add L206.  Kindergartens operate on a five-day school week.

Comments7: 397: what about the children’s socioeconomic status? – check Rzavská, D.; Masaryková, D.; Antala, B. Sustainable Development of Basic Motor Competencies Related to Socioeconomic Status of Primary School Children. Sustainability 2022, 14, 15175. https://doi.org/10.3390/su142215175

Response 7 Yes, socioeconomic status infulenced motor competence development.In our study,we mentioned “three kindergartens (sports-themed public kindergarten, ordinary public kindergarten, and inclusive private kindergarten) in the center of Haikou City”. The kindergartens are enrolled in the school by slice, and the three kindergartens all belong to the same district in the city center. Most of them from medium SES family, which we collected basic family information in another study.

Comments 8: what about the environmental impact and contribution? like in Golding J, Emmett P, Iles-Caven Y, Steer C, Lingam R. A review of environmental contributions to childhood motor skills. J Child Neurol. 2014 Nov;29(11):1531-47. doi: 10.1177/0883073813507483. Epub 2013 Oct 29. PMID: 24170258; PMCID: PMC4004720.)?

Response 8 Add L520-573

The exclusion of these environmental factors could limit the comprehensive under-standing of the observed outcomes, as they play crucial roles in shaping the overall educational experience for children in a kindergarten setting.  Future research should consider exploring these environmental variables to provide a more holistic assess-ment of the intervention's effectiveness. Another limitation was used a subjective measurement instrument, including the assessment with TGMD-3, and the monitoring of physical activity intensity during interventions, which may be influenced by indi-vidual differences among observers.

Comments 9: 404: please expand more on the conclusion, reporting the significance of the results of your research and highlight the need for further research that fills your research gaps.

Response 9. Changed

This quasi-experimental study reveals that a ball-game-based PE program is more effective in enhancing Fundamental Motor Skills (FMS) and self-regulation in kindergarten children compared to free play. The significance lies in emphasizing the potential benefits of structured PE interventions for early childhood development. Future research needs to explore the influence of teachers, facilities, and playgrounds on outcomes. Additionally, examining motor skill performance in diverse settings and establishing a comprehensive framework for acquisition can offer nuanced insights. These studies are essential to inform intervention guidelines and to promote sustainable development of children's motor skills and related areas.

The manuscript with track change is in the attachment. Thank you again for your suggestion and comments

Best regard

Authors

Reviewer 2 Report

Comments and Suggestions for Authors

Dear authors,

Firstly, I would like to congratulate the authors on the manuscript. It is very well written and organized, in addition to dealing with a topic of great relevance. Below are my specific and general comments regarding the study.

Specific comments by topics

Title:

Presents and adequately describes what will be covered in the manuscript.

Abstact:

Well structured, it describes and presents the main points that will be verified in the article.

Background:

Its structure is clearly organized, presents the theme, the study problem, as well as the justification for carrying it out.

Materials and Methods:

It presents all the important information and references for understanding what was accomplished. The treatment of groups and statistical analysis seems adequate to answer the research question.

Results:

They are organized and described clearly, allowing the reader to find the main findings of the intervention.

Recommendation:

Include captions with definitions of terms in the tables to make it easier for the reader. Although already defined in the text, it helps in the dynamics of reading and interpreting the tables. Also, indicate in the legend whether the P-value is a difference or a trend, when necessary.

Discussion:

Properly organized, presenting a relationship with other similar studies and also the authors' perspective based on the literature.

Conclusion:

Adequately points out the main message of the study in relation to what was tested and found to be the answer.

General considerations:

Although it cannot be expanded to other contexts, the findings point to important messages within the health and education contexts. It has great relevance for the areas of health and education, as it points to the importance of establishing strategies for motor development in childhood as a tool for acquiring physical literacy that will enable involvement in an active lifestyle in adolescence, in addition to the potential development of social and psychological aspects fundamental to the integral development of individuals. Additionally, it also points to the importance of physical education teachers from the initial grades to guarantee quality interventions for students.

Best regards.

Author Response

尊敬的审稿人2:

 我们谨对您花时间审阅我们的手稿表示感谢。感谢您对我们研究工作的肯定和支持。我们进一步完善了手稿,如果您有任何进一步的意见或问题,请随时告诉我们。

此致

作者

Reviewer 3 Report

Comments and Suggestions for Authors

Dear authors.

Congratulations on the work carried out, insofar as it is necessary to thank the effort involved in the development of this type of proposal. In addition, it is necessary to make some relevant considerations that influence the fact that I may consider that the work should not be published.

- Title: it does not show any reference to the sustainable development objectives that do appear in the summary, but which do not really form part of the development of the proposal, which is a major incongruity.

- Abstract: I start reading without understanding the relationship between sustainable development and the aim of the work, which is none other than to evaluate the effect of different Physical Education programmes on basic motor skills and self-regulation; furthermore, the relationship with the goals linked to sustainable development is not at all clear.

- Background: this is the introduction and there is not a single mention of the sustainable development objectives that are alluded to in the summary and which would make it possible to appreciate a link between the proposal and the characteristics of the journal.

- 2. Matherials and methods: when reference is made to the school being inclusive, is it understood that it is attended by students with disabilities? If so, I believe that the design is not correct, insofar as the differences in results that may exist between ordinary and disabled students make it impossible to establish comparisons.

- 2.1. Participants: what do you mean, at least 249? This is a scientific study, a quasi-experiment. You cannot say "at least". How many pupils exactly took part in the research? 249, which is the minimum determined by the WinPepi software; why are pupils belonging to ethnic minorities not taken into account; it is again non-specific to speak of 20-25 pupils per group.

- 2.2. Procedures: why are weight and height taken into account; I have never seen the prize as something associated with the past tests; why, could it not be something that influences pupils' motivations?

- 2.3. Intervention: when the HS group was mentioned at the beginning of the paper, there was no mention of a specific intervention. I think it would be necessary, if this is the case, to refer to the characteristics of the intervention.

- 2.3.1. HS group: it is not an objective to assess intensity through aspects such as facial colour, breathing or perspiration. Not at all; I don't see the coherence between the work on fundamental motor skills and the specific work through sport. Concretely, I think it is easier to make a design oriented to work on skills, if it is not subordinated to the technique of each sport modality.

- 2.4.1. Demographic and anthropometric data: what is the purpose of obtaining these data?

- 3. Results: I don't quite appreciate the comparisons between the intervention groups and the control group (PF). Analyses should be consistent with the design.

- Discussion: the results found are not compared with those of other research, which is what this is all about. This cannot be considered a discussion.

I believe that these considerations mean that the work should not be published in a journal such as Sustainability, either because of its content or because of the existing shortcomings.

Best regards.

Author Response

Response to Reviewer 3 Comments

Dear reviewer 3:   

We greatly appreciate you and the reviewers for your valuable feedback and for providing us with the opportunity for revisions, which allows us to further enhance our research. Due to the lack of writing experience, many details but important information in this article could not be clearly expressed. Thank you for pointing out one by one, which allowed us to make progress. Your review is like a mentor, detailed, and enlightening. We deeply feel that we are lucky to have met you and thank you for your recognition and support of our work. According to your general comments, we have created the following modification with a yellow mark in the manuscript and notes, especially, highlighted the connection of the research topic to the journal.

  1. Point-by-point response to Comments and Suggestions for Authors

Comments 1: Title: it does not show any reference to the sustainable development objectives that do appear in the summary, but which do not really form part of the development of the proposal, which is a major incongruity.

Response 1: Thank you for pointing this out. We agree with this comment. Therefore, we have changed to “Towards Sustainable Early Education Practices: A Quasi-Experimental Study on the Effects of Kindergarten Physical Education Programs on Fundamental Movement Skills and Self-Regulation from Haikou City, China”

Comments 2: - Abstract: I start reading without understanding the relationship between sustainable development and the aim of the work, which is none other than to evaluate the effect of different Physical Education programmes on basic motor skills and self-regulation; furthermore, the relationship with the goals linked to sustainable development is not at all clear.

Response 2: Agree. We have, accordingly, revised to emphasize this point.

L16-17 Quality early childhood education serves as the cornerstone for individual sustainable development and lifelong well-being. To foster the holistic development of children and establish sustainable educational practices, this quasi-experimental study, employing a pre/post-test control design, examined the influence of different kindergarten physical education programs on fundamental movement skills and self-regulation

Comments 3: - Background: this is the introduction and there is not a single mention of the sustainable development objectives that are alluded to in the summary and which would make it possible to appreciate a link between the proposal and the characteristics of the journal.

Response 3: Agree. Thanks. We add

“From the very beginning of life, early childhood care and education serve as the cornerstone for the sustained development, well-being, and health of children.  Early childhood is a crucial stage in the development of children’s physical, motor, emotion-al, and social abilities [1]. Sustainable Development Goal 4 within the Education 2030 emphasizes the need for universal access to quality early childhood development, care, and pre-primary education, aim at fostering holistic children development, ensuring that all children acquire knowledge and skills needed to promote sustainable development[2]”

Comments 4 - 2. Matherials and methods: when reference is made to the school being inclusive, is it understood that it is attended by students with disabilities? If so, I believe that the design is not correct, insofar as the differences in results that may exist between ordinary and disabled students make it impossible to establish comparisons.

Response 4  Thank you point out.

1.Maybe the problem of the translation. This kind of kindergartens in China refer to privately operated kindergartens that are certified by the education authorities, open to the public, reasonably priced, of satisfactory quality, and receive financial subsidies from the government or other forms of support https://www.hi.chinanews.com.cn/hnnew/2021-08-12/595920.html.

We change to the “non-profit private kindergarten”

  1. this study does`t include disabled students. In the L184-186 “Only healthy and typically developing children, aged 4–6 years, whose primary caregivers provided informed consent, were eligible to participate”

Comments 5 - 2.1. Participants: what do you mean, at least 249? This is a scientific study, a quasi-experiment. You cannot say "at least". How many pupils exactly took part in the research? 249, which is the minimum determined by the WinPepi software; why are pupils belonging to ethnic minorities not taken into account; it is again non-specific to speak of 20-25 pupils per group.

Response 5  The minimum determined by the WinPepi software is 249. Three kindergartens are in the urban area of the provincial capital, and we did not find any ethnic minorities in the classes. We modified the expression of class numbers:

 “Each kindergarten class was considered an experimental unit, with children from two classes each of grade K2 and K3 were selected.”

Comments 6 - 2.2. Procedures: why are weight and height taken into account; I have never seen the prize as something associated with the past tests; why, could it not be something that influences pupils' motivations?

Response 6 Thank you.1. Height and weight, as indicators of physical development, play a role in a child's overall health, and the report aims to show that children's physical development is similar in the three kindergartens.

  1. Although stickers or lollipops are commonly used as rewards in Chinese kindergartens, serving as a form of encouragement and motivation to acknowledge children's efforts and time contributed to the tests (It's not about the past test or not). There may be concerns from reviewers about the potential for triggering competitive behavior or influencing children's performance. Our method of distribution involves waiting until all the children have completed the tests, and then uniformly distributing the rewards to each child during dismissal time without prior notification.

Comments 7  - 2.3. Intervention: when the HS group was mentioned at the beginning of the paper, there was no mention of a specific intervention. I think it would be necessary, if this is the case, to refer to the characteristics of the intervention.

Response 7 : Agree, thanks. We add ball-game-based program

Comments 8 - 2.3.1. HS group: it is not an objective to assess intensity through aspects such as facial colour, breathing or perspiration. Not at all; I don't see the coherence between the work on fundamental motor skills and the specific work through sport. Concretely, I think it is easier to make a design oriented to work on skills, if it is not subordinated to the technique of each sport modality.

Response 8  thank you for your advice. While we initially considered measures of intensity, such as facial color, breathing, or perspiration, based on the reference, such as CDC mentioned talk test (https://www.cdc.gov/physicalactivity/basics/measuring/index.html

),our National Fitness Guidline (https://www.sport.gov.cn/n315/n20067006/c20324479/content.html),

and reference No.48. We acknowledge your concern that these may not be objective and accurate. Due to equipment limitations and the constraints of heart rate indicators in assessing physical activity intensity in preschool children, for example, the rapid variability and intermittent nature of physical activity intensity in preschool children, heart rate responses often lag behind changes in movement [1] . Additionally, heart rate is also significantly influenced by emotional fluctuations in children. We add perceived exertion.

L247 “simultaneously, the complexity of the motion, the applied force, and the extent of physical exertion by children also be considered”

[1] Rowlands AV, Eston RG, Ingledew DK.Measurement of physical activity in children with particular reference to the use of heart rate and pedometry [J].Spo Sport Med, 1997, 24 (4):258-272

  1. Our approach is to integrate various ball play elements to create a dynamic and engaging environment for the development of fundamental motor skills. The inclusion of ball-specific sports is designed to enhance motor skill acquisition by providing contextualized, purposeful movement experiences.

These ball-specific play settings help to form a comprehensive and integrated physical education programme that goes beyond isolated skill development and expands knowledge, skill, cognitive, social and other physical literacies.

  1. While we acknowledge the ease of designing a program focused solely on skills without tying it to specific sport modalities, our decision to include sport-specific techniques is rooted in pedagogical considerations. The integration of these ball control skills related to FMS and aims to provide children with a practical application of their motor skills in various contexts, promoting a deeper understanding and transferability of these skills.

We hope this explanation provides clarity on our program's design and its alignment with our study's objectives.

Comments 9 - 2.4.1. Demographic and anthropometric data: what is the purpose of obtaining these data?

Response 9 The basic information of the participants was provided, and the differences between the three groups of participants were compared; only height was significantly higher in the OPE group than in the other groups.

Comments 10- 3. Results: I don't quite appreciate the comparisons between the intervention groups and the control group (PF). Analyses should be consistent with the design.

Response 10 Our design had three groups. The intention was to draw meaningful contrasts between the structured physical education intervention and the unstructured free play activities in these settings. To analyze the differences among the three groups, we employed ANOVA, followed by multiple comparisons. By using multiple comparisons, we sought to identify specific nuances in the effects of the physical education curriculum in two public kindergartens compared to the free play activities in non-profit private kindergartens. We hope this clarification addresses your concerns regarding our study design and statistical methods

Comments 11 - Discussion: the results found are not compared with those of other research, which is what this is all about. This cannot be considered a discussion.

Response 11 Thank you. We acknowledge the importance of comparing our results,we are also working towards that. In general, we simplified this section by removing studies that employed different measurements as well as redundant descriptions. Instead, we focus on comparing our results with studies using methods and tools similar to those of previous investigators, thus providing additional support for our earlier findings

When we discussion the  4.1 Comparison of FMS changes in the three groups

We contrasted several interventions for structured physical activity, such as citations [56-59], to support our results. and cite the number of policies (such as [61]) have been introduced to emphasize the importance of structured physical activity. Move the citation [19]in this part.

When we discussed the 4.2 Comparison of HTKS changes in the three groups

We use the reference of [63-67], these research experimental studies surrounding the effects of various interventions on young children's Self-regulation.

Thank you for your attention to our study. We have revised the manuscript according to your suggestions, and we hope that this clarification addresses any concerns you may have had, aligning the study with the requirements of the journal.

Thank you once again for your thoughtful input. Should you have any further questions or suggestions, please feel free to let us know. We appreciate your valuable feedback.

The manuscript with track change in the attachment.

Best regards

Authors 

Reviewer 4 Report

Comments and Suggestions for Authors

Thanks for the opportunity to review this article!

The topic of the article is of interest, both for specialists and for other target groups: parents, representatives of some educational institutions. The results highlight, on the one hand, the need to introduce physical activity programs at young ages/kindergartens, on the other hand, the long-term implications of the effects that the practice of these activities bring in the development and achievement of children's subsequent performances (not only physically, but also cognitively, mentally, etc.).

The article is supported by a rich, up-to-date bibliographic list.

The design is meticulously designed and implemented.

The results are clearly presented, as well as the arguments that support the need for research.

At Discussions, I suggest that the presentation be made by discussing only comparisons with studies that used the same approach, measurement tools (the authors make a comparison with studies that used and measured other variables, even specifying this aspect - see lines 366.. .). Therefore, I suggest that this chapter be revised, the presentation of the results be in turn supported by the references to the results of previous studies (in the present form, the authors conclude the results, following that in the following sub-chapters only the results obtained by the predecessors will be presented).

In References, the source number [44] is not included in the text.

I also suggest that other limitations of the study (for example, the instruments used, etc.) be identified.

I congratulate the team of authors for the work done!

Author Response

Dear Reviewer:

We would like to express our sincere gratitude for your thorough review of our manuscript and for the positive acknowledgment of our research work. Your valuable insights and positive comments are deeply appreciated. Below I would reply to your comments and present them in track change in the manuscript.

Point-by-point response to Comments and Suggestions for Authors

Comments 1: At Discussions, I suggest that the presentation be made by discussing only comparisons with studies that used the same approach, measurement tools (the authors make a comparison with studies that used and measured other variables, even specifying this aspect - see lines 366.. .). Therefore, I suggest that this chapter be revised, the presentation of the results be in turn supported by the references to the results of previous studies (in the present form, the authors conclude the results, following that in the following sub-chapters only the results obtained by the predecessors will be presented).

Response 1: Thank you for pointing this out. We agree with this comment. Therefore, we have removed the references of studies with different measurement methods, and used the consistent methods and measurement tools of previous studies as the support for the results of this study.

Comments 2: In References, the source number [44] is not included in the text.

Response 2: Agree. We have add , now it is [52],because of the previous content increase.

 Comments 3: I also suggest that other limitations of the study (for example, the instruments used, etc.) be identified.

Response 3: yes, we add.  L553-556 “Another limitation was used a subjective measurement instrument, including the assessment with TGMD-3, and the monitoring of physical activity intensity during interventions, which may be influenced by individual differences among observers.”

Thank you once again for your encouraging remarks and for contributing to the enhancement of our manuscript. Your constructive feedback has not only validated our efforts but has also provided valuable guidance for further improvement. We are truly grateful for the time and expertise you dedicated to evaluating our work.  The manuscript with track change in the attachment

Best regards

Authors

Round 2

Reviewer 3 Report

Comments and Suggestions for Authors

Dear authors.

I greatly appreciate the work and effort you have made in trying to adapt your proposal to the considerations made in the first round of review, and, in addition, I will now detail some clarifications that I believe are very relevant in accordance with the result:

Title: that the title shows intentions linked to sustainability seems to me to be correct, as long as the content of the proposal is coherent with this change.

Abstract: the same problem of lack of connection between the proposal and sustainable development is still evident. Saying that the children's stage is relevant does not really imply anything.

Background: the link to the sustainable development goals is still not clear. Why is there no explicit reference to them?

2.2. Procedure: the inconsistency between the minimum number of participants per centre (83) and the final participants considered in the work (79, 80 and 80) has not been addressed.

2.3. Intervention: it is very inadequate to refer only to the fact that this is a balloon-based programme. Moreover, why this particular programme?

2.3.1. HS group: as already mentioned in the first review, facial colour, breathing, etc., are not objective measures related to intensity. In fact, it is not something that is considered in the web link provided by the authors. There are objective approaches that do not require many resources that allow for the assessment of physical fitness. But this is not something that can be considered; it still does not seem coherent to me, and I think it is still not justified, to use sports (specific motor skills), when the work is specifically focused on the development of basic motor skills. Not using sports does not mean a lack of context, just as their use does not guarantee such a context. It is possible to make specific and concrete use of basic motor skills outside the purely sporting sphere, which is consistent with work at this age.

Demographic and anthropometric data: it is specified that there are differences in weight, but it is not indicated how this aspect is influenced, used or considered.

3. Results: as requested in the first review, I think it is important to refer to the results obtained, in a more detailed way (comparisons between groups), regardless of the data presented in the tables.

4. Discussion: I must continue to insist on the need for the discussion to make comparisons with previous studies that have focused on the aspects analysed in the proposal. This should be the focus of the discussion.

I believe that the relationship of the proposal with the sustainable development objectives is still not fully appreciated, and therefore the presentation in this magazine is not coherent. Furthermore, there are still shortcomings that I believe should be considered as relevant.

Author Response

Dear Reviewer 3:

     Please see the attachment. Thank you for your valuable time.

Round 3

Reviewer 3 Report

Comments and Suggestions for Authors

Dear authors.

I appreciate very much the effort made in trying to adapt the proposal to what was requested in previous revisions. In addition, I would like to add some considerations that I think are relevant.

Abstract: references have been added that are still not connected in any explicit way with the sustainable development goals. This is something I said on the two previous occasions and which has not been solved despite the authors' contributions.

2.3. Intervention and 2.3.1. HS group: there is talk of a programme created by the teacher, about which not much is known, except that it focuses on sporting practices which are perhaps not the most appropriate for Physical Education at this age. Apart from this, why should a programme that only brings together "typical" proposals have a different effect or impact? I do not understand and I do not agree with this; of course, motor skills are involved when working on the sports that the authors propose, but these are the more specific motor skills, and not the basic ones, which are those that would be more appropriate, together with perceptual skills, at this age.

4. Discussion: Some improvement has been made, but indicating that the findings are coherent or are supported by other previous studies, without citing them, is not appropriate.

Regards.

Author Response

Dear Reviewer:

Best regard

Authors
